# Acute Effects of Neuromuscular Electrical Stimulation on Contralateral Plantar Flexor Neuromuscular Function

**DOI:** 10.3390/biology11111655

**Published:** 2022-11-12

**Authors:** Chris Donnelly, Timothée Popesco, Julie Rossé, Bengt Kayser, Nicola A. Maffiuletti, Nicolas Place

**Affiliations:** 1Institute of Sport Sciences, University of Lausanne, 1015 Lausanne, Switzerland; 2Human Performance Lab., Schulthess Clinic, 8008 Zurich, Switzerland

**Keywords:** maximal voluntary contraction, voluntary activation level, H reflex, V wave, coactivation, contralateral facilitation

## Abstract

**Simple Summary:**

The increase in maximal voluntary strength of one limb (e.g., right leg) while transcutaneous neuromuscular electrical stimulation is concomitantly applied to the contralateral limb (e.g., left leg) has been termed contralateral facilitation. This effect has previously been reported for the knee extensors but the underlying mechanisms are still unclear. It is also not known whether or not other muscle groups may show contralateral facilitation. Here we investigated the effect of two electrical stimulation modalities, which were compared to a submaximal voluntary contraction (~10% maximal voluntary strength) and a resting condition, on contralateral facilitation of the calf muscles. Maximal voluntary strength, and various neural parameters derived from strength and surface electromyography measurements were quantified for each condition. Our results showed that neither voluntary contraction nor electrical stimulation of the ipsilateral plantar flexors induced a contralateral facilitation of the calf muscles. This absence contrasts with the results obtained on the knee extensors and can be attributed to the absence of neural changes observed on the contralateral side. These findings should be considered by clinicians/researchers in lower-limb rehabilitation settings, as it seems easier to induce contralateral facilitation in proximal vs. distal lower limbs.

**Abstract:**

Contralateral facilitation, i.e., the increase in contralateral maximal voluntary strength that is observed when neuromuscular electrical stimulation (NMES) is applied to the ipsilateral homonymous muscle, has previously been reported for the knee extensors but the neurophysiological mechanisms remain to be investigated. The aim of this study was to compare plantar flexor contralateral facilitation between a submaximal voluntary contraction (~10% MVC torque) and two evoked contractions (conventional and wide-pulse high-frequency NMES) of the ipsilateral plantar flexors, with respect to a resting condition. Contralateral MVC torque and voluntary activation level were measured in 22 healthy participants while the ipsilateral plantar flexors were at rest, voluntarily contracted or stimulated for 15 s. Additional neurophysiological parameters (*soleus* H-reflex and V-wave amplitude and *tibialis anterior* coactivation level) were quantified in a subgroup of 12 participants. Conventional and wide-pulse high-frequency NMES of the ipsilateral plantar flexors did not induce any contralateral facilitation of maximal voluntary strength and activation with respect to the resting condition. Similarly, no alteration of neurophysiological parameters was observed in the different conditions. This absence of contralateral facilitation contrasts with some results previously obtained on the knee extensors but is consistent with the absence of neurophysiological changes on the contralateral *soleus*.

## 1. Introduction

Unilateral application of neuromuscular electrical stimulation (NMES) to the quadriceps muscle—which evokes submaximal contractions via the depolarization of motor axons [1]—may concurrently increase the maximal voluntary strength of the contralateral knee extensors with respect to a resting condition with no NMES [2,3,4]. This acute phenomenon, which can be referred to as “contralateral facilitation”, has systematically been associated to an increased level of voluntary activation (as estimated with the twitch interpolation technique), possibly indicating an increase in the efferent neural drive to the muscles promoted by NMES [3,4].

Despite the potential interest associated with the contralateral facilitation effect, these findings should be considered preliminary for several reasons. First, there is no evidence of contralateral facilitation for muscle groups other than the knee extensors. It could be hypothesized that this effect might differ for muscles with a different modulation of corticomotoneuronal/spinal excitability, such as the plantar flexors [5,6,7]. Second, the few studies having observed the contralateral facilitation effect during ipsilateral NMES used a resting condition as a comparator and force of the ‘resting’ limb was not consistently recorded [2,3,4], so it cannot be excluded that a submaximal voluntary contraction of approximately the same intensity would produce a comparable effect. Third, only one of these previous studies tested the influence of different NMES parameters on the magnitude of contralateral facilitation, but no difference was observed between a low- and a high-intensity protocol [3]. It could be hypothesized that a NMES paradigm known to induce reflexive recruitment of spinal motoneurons, such as the wide-pulse high-frequency (WPHF) modality [8,9,10], would result in greater contralateral facilitation than a conventional NMES protocol that mainly generate contractions through a peripheral mechanism (motor axon depolarization). Fourth, only global proxies of voluntary activation (twitch interpolation) and muscle excitation (surface EMG) were used in these previous studies [3,4], so that no specific neurophysiological mechanisms underlying the contralateral facilitation effect could be identified.

Therefore, the primary aim of this exploratory study was to compare the magnitude of plantar flexor contralateral facilitation (characterized by maximal voluntary strength and activation) between a submaximal voluntary contraction at 10% MVC torque and two NMES trains (conventional and WPHF, generating approximately the same force as the voluntary condition i.e., initial torque set at 10% MVC torque) of the ipsilateral homonymous muscle, with respect to a resting condition. To investigate some of the potential underlying mechanisms, the secondary aim of this study was to compare *soleus* H-reflex excitability (reflecting spinal excitability), *soleus* V-wave excitability (reflecting the level of descending voluntary drive conveyed by the motoneurons [11]), and *tibialis anterior* antagonist coactivation on the contralateral side between the four ipsilateral conditions.

## 2. Materials and Methods

### 2.1. Participants

Twenty-two healthy recreationally active participants were recruited to take part in this study. The sample size was based on previous published studies with similar aims that included 11–22 participants [2,3,4]. They were fully informed about the experimental procedures, including the risks associated with the study, before giving written informed consent to participate. Participants were young (18–45 years of age), healthy volunteers free from any neuromuscular disorders (evaluated using a standard health questionnaire), familiar with all the testing and stimulation modalities used in the study, and were requested to refrain from strenuous exercise and caffeine for 24 h before the experiment. The study was performed in accordance with the Helsinki declaration and was approved by the local ethics committee (Commission d’éthique de la recherche sur l’être humain du Canton de Vaud, no. 2016-00767).

### 2.2. Experimental Approach

Each participant took part in a single experimental session that consisted of a preparation phase—during which subjects were positioned and individual stimulation intensities were carefully determined [10,12,13]—and an experimental phase (Figure 1), during which the ipsilateral plantar flexors were subjected to four different experimental conditions while the contralateral plantar flexors were concurrently evaluated with tibial nerve stimulation, surface EMG, and maximal voluntary contractions (MVC). The experimental conditions were passive rest (REST), a submaximal voluntary contraction (VOL), a conventional NMES train (CONV) and a WPHF NMES train (WPHF), all lasting 15 s and generating approximately the same initial torque (10% MVC), except REST. The primary outcomes were contralateral MVC torque and voluntary activation level. The secondary outcomes were contralateral *soleus* H-reflex and V-wave amplitude as well as *tibialis anterior* coactivation level. Primary outcomes were obtained from 22 subjects (4 women: mean ± SD, 26 ± 4 yrs, 162 ± 7 cm, 65 ± 9 kg and 18 men: 26 ± 6 yrs, 180 ± 5 cm, 76 ± 6 kg), while EMG-based secondary outcomes from a subgroup of 12 participants (4 women and 8 men: 28 ± 8 yrs, 179 ± 7 cm, 76 ± 5 kg).

### 2.3. Experimental Setup

#### 2.3.1. Torque Recordings

Ipsilateral and contralateral plantar flexion torque was recorded with two instrumented pedals equipped with a strain gauge sensor (capacity: 110 N.m, Vishay Micro Measure, Raleigh, NC, USA). Participants were seated with joint angles of 90° at the hips, knees, and ankles. Their feet were securely strapped to the pedal with Velcro straps and their thighs were also clamped down to the pedal. The torque signal was recorded at 1250 Hz using an analog-to-digital converter (MP150, Biopac Systems Inc., Goleta, CA, USA).

#### 2.3.2. EMG Recordings

Surface EMG signals of the contralateral *soleus* and *tibialis anterior* muscles were recorded with pairs of circular silver-chloride electrodes (recording diameter: 1 cm; inter-electrode distance: 2 cm) (Meditrace 100, Tyco, Markham, Canada) that were placed lengthwise over respective muscle bellies according to SENIAM recommendations [14]. The reference electrode was placed on the ipsilateral patella [10,12,13]. Inter-electrode resistance was reduced by shaving and cleaning the skin with alcohol. EMG signals were amplified (×1000), band-pass filtered (10–500 Hz), digitized at a sampling frequency of 5 kHz, and recorded with the analog-to-digital converter (MP150, BIOPAC, Goleta, CA, USA). Torque and EMG data were stored and later analyzed with a commercially available software (Acqknowledge version 4.2, Biopac Systems Inc., Goleta, CA, USA).

#### 2.3.3. Tibial Nerve Stimulation

The contralateral tibial nerve was stimulated with (1) submaximal single pulses—to evoke the largest H-reflex response (Hmax) at rest, (2) supramaximal single pulses—to evoke the largest M-wave response at rest (Mmax) and during the MVC (superimposed M wave, Msup), as well as the V-wave response (V) obtained during the MVC, and (3) supramaximal paired pulses during and after the MVC—to estimate the voluntary activation level according to the twitch interpolation technique. All stimuli were delivered by a cathode electrode (diameter: 1 cm) placed in the popliteal fossa over the tibial nerve and an anode (10 × 5 cm, Compex, Ecublens, Switzerland) positioned 2–3 cm proximal to the patella [10,12,13]. Rectangular pulses (duration: 1 ms) were generated by a high-voltage (max: 400 V) constant-current stimulator (Digitimer DS7AH, Hertfordshire, UK).

#### 2.3.4. NMES

A second programmable constant-current stimulator (Digitimer DS7AH, Hertfordshire, UK) was used to deliver conventional and WPHF NMES to the ipsilateral calf muscles. Two rectangular (10 × 5 cm) self-adhesive electrodes (Uni-Patch, Wabasha, MN, USA) were used; the proximal electrode was positioned over the muscle belly of both gastrocnemii (~4 cm below the popliteal fossa) and the distal one over the *soleus* muscle belly (~10 cm above the calcaneus) [10,12,13]. Pulse duration and frequency were, respectively, 0.1 ms and 30 Hz for conventional NMES and 1 ms and 100 Hz for WPHF NMES [8,10,15,16].

### 2.4. Experimental Protocol

Most of the experimental procedures described below have also been detailed in our previous works [10,12,13].

#### 2.4.1. Preparation Phase

This phase started with contralateral tibial nerve stimulation at rest to determine the optimal submaximal intensity for evoking Hmax as well as the supramaximal intensity. Single pulses of progressively increasing intensity were delivered every 8 s, starting from 5 mA. The increments were adjusted on an individual basis (typically by 1–2 mA) during the acquisition, with particular care around the Hmax response. The individual current intensity associated to Hmax (30 ± 12 mA on average) was to be consistently used during the experimental phase in all the conditions. Then, stimulation intensity of single pulses was progressively increased (typically by 10–20 mA) until plateaus in *soleus* Mmax and peak twitch torque were observed. This intensity was further increased by 20% to ensure stimulus supramaximality (92 ± 24 mA), which was then to be consistently used for single and double pulses delivered during the experimental phase.

Subsequently, we measured ipsilateral plantar flexion MVC torque to define the 10% MVC torque target. Participants warmed up by performing 8–10 submaximal contractions between 20% and 80% of their estimated MVC torque. They were then asked to perform 2–3 MVCs of ~5 s (1–2 s to progressively build up force), with rest periods of 1 min between each trial. Once the 10% MVC torque target was calculated (13 ± 3 N.m), we delivered multiple 1 s NMES trains to determine the current intensity that evoked a torque level as close as possible to the 10% MVC target. This procedure was first conducted for WPHF and then for CONV NMES, and the respective stimulation intensities (20 ± 7 and 74 ± 13 mA) were then consistently used during the experimental phase.

The last part of the preparation phase consisted in a standardized warm up of the contralateral plantar flexor and dorsiflexor muscles (8–10 submaximal contractions between 20% and 80% of their estimated MVC torque), followed by the completion of 2–3 MVCs of the dorsiflexors separated by 1 min. These contractions were conducted to record the maximal EMG activity of the *tibialis anterior* muscle that was subsequently used to calculate its level of antagonist coactivation (i.e., during a MVC of the plantar flexors).

#### 2.4.2. Experimental Phase

A schematic overview of the experimental phase is shown in Figure 1. For the four different experimental conditions (REST, VOL, CONV, and WPHF), participants were asked to fully relax the ipsilateral plantar flexors (REST), to perform a sustained voluntary contraction with a visual feedback (horizontal line) showing the 10% MVC torque target (VOL), and to not voluntarily contract while receiving NMES trains at the pre-determined stimulation intensities for both CONV and WPHF. These intensities were carefully re-verified and sometimes slightly readjusted 1 min before the start of each NMES condition using a test train of 1–2 s, always with the goal to generate an initial torque level of 10% MVC. During each of these 15 s phases, the contralateral tibial nerve was stimulated at rest (first 5 s) and then during and after a 5 s MVC to evoke the following responses: (1) *soleus* Hmax at 1 s, (2) *soleus* Mmax at 3–4 s, (3) superimposed doublet at 6–7 s, (4) *soleus* Msup and V at 8–9 s, and (5) resting potentiated doublet at 11–12 s. The four experimental conditions were presented randomly (two trials per condition separated by 1 min) and interspersed by 3 min-long rest periods.

### 2.5. Data Analysis

The MVC torque was considered as the highest torque (single data point) voluntarily attained during the MVC. Only the trial with the highest MVC torque recorded in each condition was further considered. Voluntary activation level was quantified using the following formula: (1 − (superimposed doublet torque × (torque level at stimulation/MVC torque)/potentiated doublet torque)) × 100 [17]. The peak-to-peak amplitudes of the different EMG responses (*soleus* Hmax, Mmax, Msup, and V) were quantified and the resultant Hmax/Mmax as well as V/Msup ratios were calculated. *Tibialis anterior* coactivation level was calculated by normalizing its EMG activity (root mean square amplitude over a 500 ms interval) recorded during plantar flexion MVC, to the maximal EMG activity obtained during dorsiflexion MVC, as a percentage [18,19]. We also quantified the average ipsilateral submaximal torque generated in each experimental condition (mean of 15 s) and expressed it relative to the respective MVC torque. The ipsilateral torque when a stimulation was sent to the contralateral tibial nerve was also quantified. This was also expressed relative to MVC torque and the data are shown in Appendix A. Participants generating an increasing torque (i.e., a positive extra torque value) during the stimulation in WPHF were considered as responders in accordance with previous studies [10,13].

### 2.6. Statistical Analysis

Data were plotted in graphical format to assess sampling distributions using GraphPad Prism (GraphPad Inc., San Diego, CA, USA). A one-way repeated measures ANOVA was used to test for potential differences in contralateral MVC torque, *soleus* Hmax/Mmax ratio, *soleus* V/Msup ratio between the four experimental conditions (REST, VOL, CONV, and WPHF). In case of a significant main effect, Tukey post-hoc tests for multiple comparisons were used. Effect sizes for each ANOVA were calculated as partial eta squares (η_p_^2^) and Cohen’s d for post hocs. A Friedman’s ANOVA was used to test for potential differences in contralateral voluntary activation level, ipsilateral average submaximal torque, ipsilateral torque at stimulation, and *tibialis anterior* coactivation level between conditions (REST, VOL, CONV, and WPHF), as these variables were not normally distributed. A t-test was used to test for differences between responders and non-responders to WPHF. For t-tests the effect size was calculated as Cohen’s d. Spearman’s correlation coefficient (r_S_) was used to assess associations between pairs of variables. Data were analyzed using Jamovi (version 0.9.6.9, Jamovi software, Sydney, Australia) and power analyses were calculated using G*Power software (version 3.1.9.4, G*Power software, Düsseldorf, Germany). The level of significance was set at *p* ≤ 0.05.

## 3. Results

### 3.1. Contralateral Side

A significant main effect of condition was observed for contralateral MVC torque (*p* = 0.022, 1-β = 0.933, η_p_^2^ = 0.147) but not for voluntary activation level (*p* = 0.92). MVC torque was significantly lower for VOL than for CONV (*p* = 0.042, Cohen’s d = 0.611), with no other difference between the remaining conditions (Figure 2A). Voluntary activation level did not differ significantly between the four experimental conditions (Figure 2B). A significant main effect of condition was observed for *soleus* Hmax/Mmax ratio (*p* = 0.026, η_p_^2^ = 0.360), but post hoc comparisons revealed no significant difference between the four experimental conditions (Figure 3A). No main effect of condition was observed for *soleus* V/Msup ratio (*p* = 0.60, η_p_^2^ = 0.056) (Figure 3B) and for *tibialis anterior* coactivation level (*p* = 0.92) (Figure 3C).

### 3.2. Ipsilateral Side

The mean plantar flexion MVC torque was 128 ± 33 N.m. The average submaximal torque recorded during the 15 s experimental phases was 4.4% (range: 1.0–6.5%), 10.7% (range: 9.3–11.8%), 11.6% (range: 9.8–15.9%), and 16.2% (range: 11.2–22.9%) of MVC torque for REST, VOL, CONV, and WPHF (representative traces are presented in Figure 4), respectively (median and interquartile range), with a significant main effect of condition (*p* < 0.001). Submaximal torque was significantly lower for REST compared with all the other conditions (*p* < 0.001). Submaximal torque of CONV was also significantly higher than VOL (*p* = 0.045) and submaximal torque of WPHF was higher compared with both VOL (*p* < 0.001) and CONV (*p* = 0.004). The mean-evoked torque was higher in responders to WPHF compared with non-responders (26 ± 12% vs 13 ± 5% MVC torque; *p* = 0.004, Cohen’s d = 1.39) but this did not influence the parameters recorded on the contralateral side.

## 4. Discussion

The main findings of this exploratory study were that conventional and wide-pulse NMES of the ipsilateral plantar flexors did not induce any contralateral facilitation of maximal voluntary strength and activation with respect to a resting condition, while an ipsilateral voluntary contraction of the same intensity resulted in lower contralateral strength than conventional NMES. Contralateral *soleus* H-reflex and V-wave excitability as well as *tibialis anterior* coactivation were not influenced by the different ipsilateral conditions.

Contrary to previous findings obtained on the knee extensors [2,3,4], NMES of the plantar flexor muscles did not induce any contralateral facilitation effect in this study, as both MVC torque and voluntary activation level were basically unchanged during conventional NMES, wide-pulse NMES and the REST condition. Even if our current study was not designed to investigate potential differences in contralateral facilitation between different muscle groups, and the experimental conditions were not exactly the same in these different studies, it is tempting to speculate that the contralateral facilitation effect may be easier to induce on the knee extensors [2,3,4] than on the plantar flexors. This could be due to a different modulation of corticospinal/spinal excitability, as evidenced by the different behavior of (cervicomedullary) motor evoked potentials between these two muscles at high force levels [5,6,7].

Contralateral MVC torque was lower for VOL than for CONV NMES—despite comparable though significantly different contraction intensities (10.7 vs 11.6% MVC torque, respectively)—thereby showing that light voluntary contractions of the plantar flexors did not result in a contralateral facilitation effect and even inhibited maximal voluntary strength compared to conventional NMES. It is difficult to discuss these results as none of the previous contralateral facilitation studies compared the effect of ipsilateral NMES to voluntary contractions of comparable intensity [2,3,4]. What is, however, well known is that motor unit recruitment patterns are considerably different between electrically evoked and volitional contractions of the same intensity, with a spatially fixed, temporally synchronous, and nonselective pattern of activation (with a preferential recruitment of fast units) imposed by NMES [20]. However, how this specificity and all the associated physiological consequences—such as the exaggerated metabolic cost of NMES-induced contractions [21,22,23]—may have impacted the present contralateral facilitation results is difficult to infer and worthy further investigation.

Contrary to one of our hypotheses, WPHF did not produce a greater contralateral facilitation effect than CONV NMES. WPHF has been shown to have a major effect on ipsilateral motoneurons, as the progressive increase in force during the stimulation is attributed to reflexive recruitment of MUs [9,24] presumably through to a combination of temporal summation of post-synaptic potentials [25] and increased persistent inward currents [8,10,26,27]. In the present study, 12 out of 22 participants showed an increased torque during WPHF (i.e., ‘responders’), consistent with the 40–60% of responders to WPHF reported in previous studies [10,12,15]. This resulted in a higher evoked torque response (~20% MVC torque) as compared with the initial 10% MVC target torque, but this was apparently not enough to significantly alter the neural properties of the contralateral plantar flexors. Further analysis of the responders vs. non-responders results showed no subgroup differences for the main outcomes of this study, which is the reason why we did not focus on this aspect in the results. These analyses do not provide any evidence supporting of the use of ipsilateral NMES in a clinical setting to enhance contralateral plantar flexor muscle function albeit our small sample size limits extrapolation of findings to larger populations. Although there was no significant change in neural properties in response to WPHF at the group level, the change in MVC torque (range: −13.51 to +20.99%) correlated with the change in voluntary activation level (range: −15.18 to +13.52%) (r_S_ = 0.72, *p* < 0.001), while no such association was found when a contralateral facilitation was reported [3]. It should also be noted that this correlation was previously found in knee extensors despite the absence of change in MVC torque at the group level [4]. As this correlation was only found in the WPHF condition in our study, we speculate that there might be greater potential for WPHF than CONV to induce contralateral facilitation of the plantar flexors, but this remains to be tested using different experimental paradigms.

One of the originalities of the present work consists in the use of neurophysiological techniques to assess the mechanisms underlying the potential contralateral facilitation effect, which were not investigated in the previous studies conducted on the knee extensors (in which contralateral facilitation was observed) [2,3,4]. In these previous works, only the maximal voluntary EMG activity was quantified and it was found to increase at group mean level in the contralateral *vastus lateralis* and/or *rectus femoris* muscles [3,4], although not consistently [2]. The Hmax/Mmax and V/Msup ratios, respectively used as indexes of spinal excitability and descending voluntary drive to the motoneurons, did not differ between the various experimental conditions. While we are not aware of any study with V-wave measurement to investigate the contralateral facilitation effect, Milosevic et al. [28] reported no changes in transcutaneous spinal cord stimulation evoked spinal reflexes in the contralateral *soleus* after the application of NMES for 60 s (pulse width of 300 μs, frequency of 40 Hz, stimulation intensity of 150% motor threshold, no information about the evoked force level) or force-matched voluntary contractions of the ipsilateral plantar flexors. Kato et al. [29] stimulated the median nerve for 70 s (pulse width of 400 μs, frequency of 20 Hz) to elicit a wrist flexion of 10% MVC force and observed contralateral facilitation of transcutaneous spinal cord stimulation evoked spinal reflexes in thigh muscles (*vastus medialis* and *biceps femoris*) but not in the *soleus*. Mendonca et al. [30] failed to report a change in the contralateral MVC force of the plantar flexors after four weeks of unilateral resistance training; interestingly they also observed unchanged Hmax/Mmax and V/Msup ratios in the *soleus*. Altogether, these studies provide additional evidence for the higher susceptibility of knee extensors vs. plantar flexors to contralateral facilitation. Besides, our results showed no change in *tibialis anterior* coactivation level. Cattagni et al. [3] reported an increased coactivation level accompanying the increased MVC force production on the contralateral knee extensors when NMES elicited 30% MVC torque—but not at a lower stimulation intensity eliciting 10% MVC torque. This increased coactivation paralleled the *rectus femoris* agonist EMG activity and thus could not explain the increased net knee extension torque. Together, these results indicate that the investigated neural properties of the plantar flexor muscles were not affected by contralateral NMES and thus our findings do not shed light on possible mechanisms underpinning the contralateral facilitation effect.

The present study is not exempt from limitations. The relatively low torque level evoked on the ipsilateral limb (~10% MVC), which was necessary to limit the potential antidromic collision in response to WPHF [31], might have been too low to elicit a contralateral facilitation in the plantar flexors, although such a contraction intensity was found to be sufficient in the larger knee extensors [3]. In addition, plantar flexor EMG activity was not considered for the *gastrocnemii* muscles but only for the *soleus* muscle, which is the main contributor to plantar flexion torque at a knee angle of 90° [32].

## 5. Conclusions

To conclude, and contrary to our initial hypothesis, there was no increase in MVC torque of the contralateral plantar flexors while the ipsilateral plantar flexors were stimulated with CONV, WPHF, or were voluntarily contracted to reach 10% of MVC torque. This absence of contralateral facilitation contrasts with results obtained on the knee extensors but is confirmed by the absence of neurophysiological changes observed on the contralateral *soleus*. From a practical perspective, the findings of the present study suggest that—contrary to the knee extensors—unilateral NMES of the plantar flexors induce no contralateral facilitation effect. This new information may be useful to clinicians/researchers in rehabilitation/laboratory settings.

## Figures and Tables

**Figure 1 biology-11-01655-f001:**
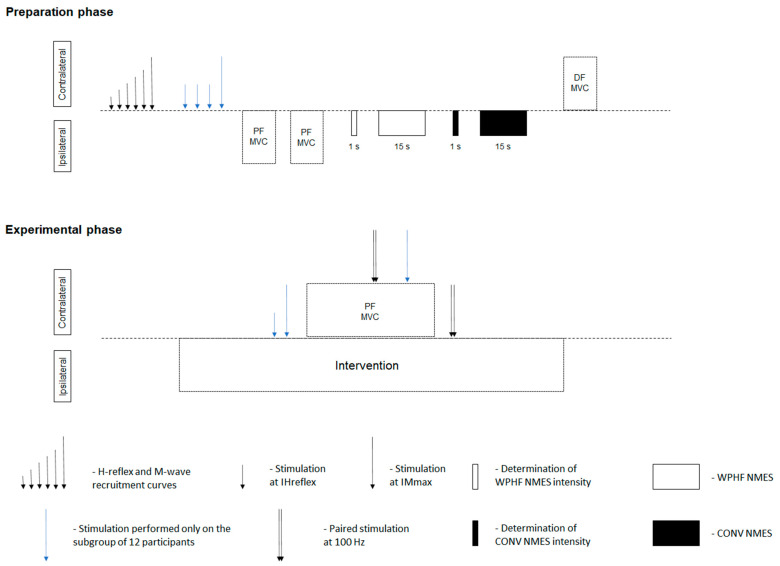
Schematic diagram of the experimental design. IHreflex = stimulation intensity used to elicit a maximal H-reflex. IMmax = 120% of stimulation intensity used to elicit a maximal M-wave. PF MVC = plantar flexor maximal voluntary contraction. Stimulation intensity of wide-pulse, high-frequency neuromuscular electrical stimulation (WPHF NMES, stimulation frequency: 100 Hz, pulse duration: 1 ms) and conventional (CONV) NMES (stimulation frequency: 30 Hz, pulse duration: 0.1 ms) was set to produce an initial force level of 10% maximal voluntary contraction torque. DF MVC = dorsiflexor maximal voluntary contraction. The 15 s intervention on the left leg was either rest (REST), a voluntary contraction at 10% MVC torque (VOL), and conventional (CONV) or wide-pulse high-frequency (WPHF) NMES. The PF MVC during the intervention phase lasted for 5-s. EMG signals of the *soleus* muscle were recorded throughout the experiment on a subgroup of 12 participants. Blue arrows represent stimulations performed exclusively on the subgroup of 12 participants.

**Figure 2 biology-11-01655-f002:**
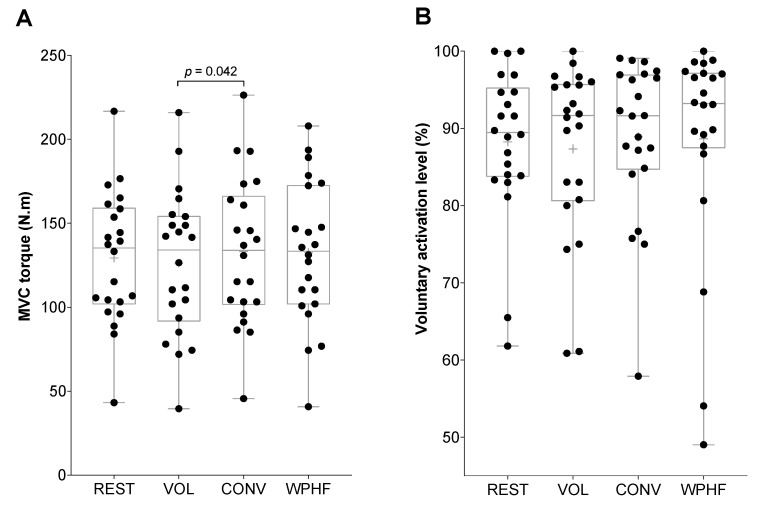
Right leg plantar flexor (**A**) MVC torque while the left plantar flexors were at rest (REST), contracted at 10% MVC torque (VOL) or stimulated with conventional NMES (CONV) or wide-pulse high-frequency NMES (WPHF) and (**B**) right leg plantar flexor voluntary activation level in REST, VOL, CONV, and WPHF conditions. Individual data are presented as dots on a boxplot. The horizontal lines on the box represent the 75th, 50th (median) and 25th percentiles and the cross represents the mean. The whiskers extend to the maximum and minimum values. Only the *p*-value of a significant post-hoc comparison is shown on the graph.

**Figure 3 biology-11-01655-f003:**
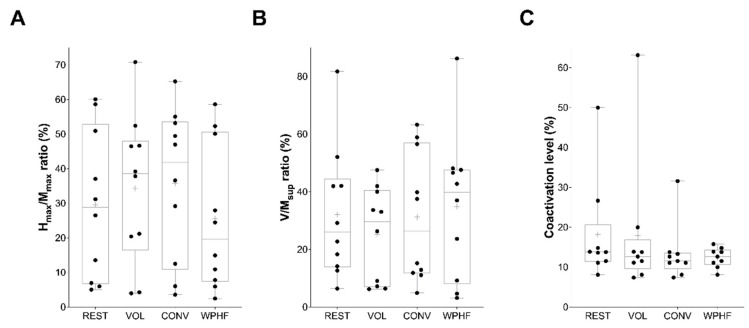
Right leg (**A**) *soleus* Hmax/Mmax ratio, (**B**) *soleus* V/Msup ratio and (**C**) *tibialis anterior* coactivation level while the left plantar flexors were at rest (REST), contracted at 10% MVC torque (VOL) or stimulated with conventional NMES (CONV) or wide-pulse high-frequency NMES (WPHF). Data of the subgroup of 12 participants for whom EMG signals were recorded are presented. Individual data are presented as dots on a boxplot. The horizontal lines on the box represent the 75th, 50th (median), and 25th percentiles and the cross represents the mean. The whiskers extend to the maximum and minimum values.

**Figure 4 biology-11-01655-f004:**
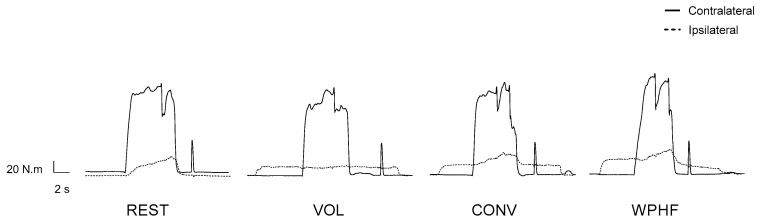
Plantar flexor torque recordings from one participant. The torque–time traces of the contralateral (full line) and ipsilateral (dot line) plantar flexors are presented for the four conditions investigated: REST: Rest; VOL: voluntary contraction at 10% MVC torque; CONV: conventional NMES; WPHF: wide-pulse high-frequency NMES.

## Data Availability

Data are available from the corresponding author upon reasonable request.

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
