# Peer review of "Acute Effects of Neuromuscular Electrical Stimulation on Contralateral Plantar Flexor Neuromuscular Function"

_biology, 2022, doi:10.3390/biology11111655_

Round 1

Reviewer 1 Report

Dear Author,

In this article, you investigated the acute effects of ipsilateral contraction of plantar flexor muscles on contralateral limb function. It could be an interesting article; however, the conclusions did not confirm the initial hypothesis. In these cases, it is highly recommended a rigorous methodological approach. Methods section is well described, on the other hand the limited sample could affect the results. There is no mention of the sample calculation, or size effect. 

Line 1. “Type of the Paper (Article, Review, Communication, etc.)”. Please specify that it is an article. 

TITLE:

“Acute effects of neuromuscular electrical stimulation on contralateral plantar flexor neuromuscular function.” You considered other conditions as 10% MVC torque. The title should be rephrased. 

INTRODUCTION:

“generating approximately the same force as the voluntary condition”. Please quantify the “approximately”. In a quantitative study, the methodological approach is fundamental. 

METHODS

Please specify the sample calculation and the sample size.

There is no mention on the characteristic of the investigators involved in the experimental phase. It is highly recommended a previous training to place the sEMG electrodes. Moreover, if more than one investigator (recommended) placed the electrodes, it is recommended an inter-rater evaluation.  

“The reference electrode was placed on the ipsilateral patella”. Please explain or add a reference.

“Tibial nerve stimulation”. Please add a reference.

“NMES”. Please add a reference to justify the electrodes positioning. 

There is no reference on the preparation and experimental phase. It is a new approach or already described in literature?

RESULTS

I suggest to improve results section by indicating the values obtained in the experimental phase. By the graph, it seems that there is a high heterogeneity in the obtained values (high interquartile range).

Reviewer 2 Report

Dear editors and authors,

I express my gratitude for the opportunity to review this manuscript for possible publication.
In this study the authors examine the acute effects of neuromuscular electrical stimulation on contra-lateral plantar flexor neuromuscular function.

Although the study has the potentiality of being shared with the scientific community, I believe that the manuscript would benefit from a minor revision with the attempt to better support their experimental setting.

Therefore, I hope to help authors improve the structure and content of the manuscript.
Although the content is well argumented, I would invite the authors to focus attention on the following:

1.               Abstract. The authors should start with a short intro that better highlights their work and end up with a paragraph that include results.

2.               The theoretical framework is scarce, they should clearly describe the scientific evidence that supports the hypothesis they have raised.

3.               The authors should define better inclusion and exclusion criteria

4.               I would like to see more of the practical implications. Based on the analyzed variables, how the authors intend to use their findings? What can be done to improve the students’ resilience with a view to attaining academic achievement?

kind regards

Reviewer 3 Report

The article “Acute effects of neuromuscular electrical stimulation on contra-lateral plantar flexor neuromuscular function” shows no positive results in distal hindlimbs muscles. It is a genuine interest article, excellently written and explained.

 A minor problem with the articles is the need of more clear results to ensure the lack of effect. So, it is important to include the effect size of the ANOVAs (np2) and for any post hoc significant result.

 Moreover, it is important to include the power (1-β err prob) of the ANOVAs, to ensure the lack of results. The authors can use the G-Power software to calculate the power of the analysis in a Post hoc calculation of the results.

 In addition, in the discussion (line 320), the authors discuss the responders and non-responders. Can the authors include one result to speak about the presence (or not) of responders and non-responders?

 Finally, in line 266, the “3.2 Ipsilateral side” results are graphically presented with representative traces. Can the authors include a result graph like Figures 2 and 3 with the data?

Round 2

Reviewer 1 Report

Dear Author,

Thank you for addressing my suggestions. The article has been improved but I still have some concerns about methods.

Inclusion criteria:

“Participants were young, healthy volunteers free from any neuromuscular disorders”

Please quantify young, and describe the concept of healthy. I suggest: the inclusion criteria were: (I) ..-.. years old; (II)….; (III)….

“sEMG recordings”

Please give a more detailed description of the data analysis. Did you calculate RMS, peak amplitude? Which was the sampling frequency of the sEMG? How did you menage lack of signal? Please provide the formula to calculate MVC (RMS/RMS MVC?)

If I could suggest, please refer to:

Demeco A, et al. Quantitative analysis of movements in facial nerve palsy with surface electromyography and kinematic analysis. J Electromyogr Kinesiol. 2021 Feb;56:102485. doi: 10.1016/j.jelekin.2020.

Figure 2 B

After rest 2 people obtained a 60-70% of the activation % of voluntary activation level. How do you explain this?

Limitations:

Please include the sample population limiting the generalizability of the conclusions.
